# Novel Markers of the Metabolic Impact of Exogenous Retinoic Acid with A Focus on Acylcarnitines and Amino Acids

**DOI:** 10.3390/ijms20153640

**Published:** 2019-07-25

**Authors:** Joan Ribot, Andrea Arreguín, Ondrej Kuda, Jan Kopecky, Andreu Palou, Maria Luisa Bonet

**Affiliations:** 1Grup de Recerca Nutrigenòmica i Obesitat, Laboratori de Biologia Molecular, Nutrició i Biotecnologia (LBNB), Universitat de les Illes Balears, 07122 Palma de Mallorca, Spain; 2CIBER de Fisiopatología de la Obesidad y Nutrición (CIBEROBN), 28029 Madrid, Spain; 3Institut d’Investigació Sanitària Illes Balears (IdISBa), 07120 Palma de Mallorca, Spain; 4Department of Adipose Tissue Biology, Institute of Physiology of the Czech Academy of Sciences, 14220 Prague 4, Czech Republic

**Keywords:** acylcarnitines, amino acids, targeted metabolomics, retinoic acid, vitamin A

## Abstract

Treatment with all-trans retinoic acid (ATRA), the carboxylic form of vitamin A, lowers body weight in rodents by promoting oxidative metabolism in multiple tissues including white and brown adipose tissues. We aimed to identify novel markers of the metabolic impact of ATRA through targeted blood metabolomics analyses, with a focus on acylcarnitines and amino acids. Blood was obtained from mice treated with a high ATRA dose (50 mg/kg body weight/day, subcutaneous injection) or placebo (controls) during the 4 days preceding collection. LC-MS/MS analyses with a focus on acylcarnitines and amino acids were conducted on plasma and PBMC. Main results showed that, relative to controls, ATRA-treated mice had in plasma: increased levels of carnitine, acetylcarnitine, and longer acylcarnitine species; decreased levels of citrulline, and increased global arginine bioavailability ratio for nitric oxide synthesis; increased levels of creatine, taurine and docosahexaenoic acid; and a decreased n-6/n-3 polyunsaturated fatty acids ratio. While some of these features likely reflect the stimulation of lipid mobilization and oxidation promoted by ATRA treatment systemically, other may also play a causal role underlying ATRA actions. The results connect ATRA to specific nutrition-modulated biochemical pathways, and suggest novel mechanisms of action of vitamin A-derived retinoic acid on metabolic health.

## 1. Introduction

Substrate oxidation coupled to the generation of either ATP or heat opposes body fat gain and contributes to decreased levels of circulating lipids and glucose. Agents enhancing oxidative metabolism in mammalian tissues are therefore of potential value in the management of obesity, hyperlipidemia and diabetes. To advance in the identification and evaluation of such agents, readily available markers from easily accessible biological samples are required that reflect the response to the intervention in less accessible tissues. Furthermore, knowledge on these markers should contribute to a better understanding of the agent’s mechanisms of action and, ultimately, the control of mammalian energy metabolism and its relationship with intermediary metabolism. Plasma and blood cells are potential sources of such biomarkers [1,2].

All-trans retinoic acid (ATRA), the main carboxylic form of vitamin A (retinol), stimulates uncoupling protein 1 (UCP1)-dependent thermogenesis in brown adipose tissue (BAT) [3,4,5] and fatty acid (FA) oxidation (FAO) and mitochondrial oxidative metabolism in white adipose tissue (WAT) [6,7,8], skeletal muscle [9,10,11], and the liver [12,13,14,15]. These activities were initially demonstrated in intact rodents and model cells treated with exogenous ATRA, and multiple lines of evidence suggest that endogenous ATRA produced intracellularly from vitamin A may have similar effects. Dietary vitamin A deficiency results in mice in increased body fat and decreased BAT UCP1 content, which are both reversed by ATRA treatment [5,16]. Lower ATRA levels in BAT are linked to defects in cold-induced thermogenesis in lipocalin 2 knockout mice [17]. Genetic ablation of retinol dehydrogenases involved in ATRA synthesis results in increased adiposity and defects in BAT thermogenesis in mice [18,19]. Further, mice with blunted retinoic acid receptor (RAR) signaling in the liver—owing to liver-specific transgenic expression of a dominant negative form of RARα—show decreased hepatic FAO and develop liver steatosis [20]. ATRA effects on lipid and energy metabolism are mediated by multiple mechanisms, including the activation of specific nuclear receptors (notably the canonical RARs and the β/δ isoform of peroxisome proliferator-activated receptor, PPAR β/δ), protein kinases, and hormonal pathways [10,11]. Importantly, both in lean and obese ATRA-treated mice, decreases in body weight and adiposity and improvements in glucose tolerance, insulin sensitivity, blood lipids, and liver lipid content are observed [6,10,12,21,22]. 

We here sought to get further insight into the mechanisms of ATRA metabolic action, and to identify/validate blood markers of increased tissue oxidative/thermogenic activity and metabolic health, by exploiting the model of exogenous ATRA treatment in mice. To this end, we conducted targeted metabolomics analyses of plasma and peripheral blood mononuclear cells (PBMC) comparatively in mice treated with ATRA or placebo, using high-throughput mass spectrometry-based screening, primarily of acylcarnitines (ACs) and amino acids (AA), in an extension of a technique normally used for the screening of metabolic disorders in human newborns. ACs are intermediate metabolites in the oxidation of FA and AAs that comprise an acyl group esterified to an L-carnitine molecule [23,24]. Plasma AC levels have attracted attention as markers of obesity-associated metabolic disturbances such as insulin resistance, yet this relationship is complex [24]. Increased levels of long-chain ACs in plasma and tissues have been described in association with obesity and insulin resistance [25,26,27,28,29], but also after weight loss and exercise paralleling increases in systemic insulin sensitivity [30,31,32]. Similar to ACs, AAs such as branched-chain AAs and aromatic AAs have been proposed as potential markers for obesity-related metabolic disturbances on the basis of metabolomics studies [33,34].

## 2. Results 

We applied a short-term, high-dose ATRA treatment protocol previously developed and used in our laboratory to study ATRA effects on lipid and energy metabolism in mice [6,8,9,16,21]. The treatment triggered as expected a decrease in body weight independent of significant changes in food intake, resulting in lower energy efficiency and a decreased adiposity index in the ATRA group (Appendix A), consistent with enhanced lipid mobilization and energy metabolism.

### 2.1. Plasma Analyses 

#### 2.1.1. Plasma Acylcarnitine Profile 

AC profiling showed increased levels of most AC species in plasma of ATRA-treated mice compared to controls, including short- (C2-C7), medium- (C8-C14), and long-chain (C16-C20) AC species (Figure 1). The increment reached statistical significance after Benjamini–Hochberg correction for C2:0 (acetylcarnitine, 74% increase), C4OH (80% increase), C8:0 (72% increase), C12:1 (137% increase), and C16DC (47% increase) ACs. Plasma levels of free carnitine were also significantly higher (by 53%) in the ATRA-treated mice. 

#### 2.1.2. Plasma Amino Acid Profile

Relative to the control mice, the ATRA-treated mice had equal plasma arginine levels but significantly decreased plasma citrulline levels (by 31%) (Figure 2). The “global arginine bioavailability ratio” (GABR), defined as the ratio of plasma arginine to ornithine plus citrulline [35,36], was calculated to be 35% higher in the ATRA group (*p* < 0.001, two-tailed Student’s *t* test). The GABR is a reliable approximation of nitric oxide (NO) synthetic capacity in vivo, as it reflects the systemic bioavailability of arginine for NO biosynthesis better than plasma arginine levels [35,36]. ATRA-treated mice also had trends toward increased plasma creatine (↑31%) and taurine levels (~7-fold increase) relative to controls (Figure 2). The sum of taurine and glycine levels was higher in ATRA mice than in controls ([taurine + glycine]: 317 ± 108 vs. 100 ± 11 arbitrary units/volume in %, *p* = 0.050, two-tailed Student’s *t* test). ATRA-treated mice had higher cystine levels but equal levels of homocysteine and homocystine in plasma compared to control mice. Likewise, plasma levels of other AAs shown in Figure 2 were unaffected by ATRA treatment, including branched chain (leucine, isoleucine, and valine) and aromatic AAs (phenylalanine, tryptophan, and tyrosine). 

#### 2.1.3. Plasma Levels of Targeted Lipids

ATRA-treated mice had plasma levels of the n-3 polyunsaturated FA (PUFA) docosahexaenoic acid (DHA, 22:6n-3) significantly increased by 63% compared to controls, as well as trends toward increased plasma levels of eicosapentaenoic acid (EPA, 20:5n-3; ↑29%) and arachidonic acid (ARA, 20:4n-6; ↑35%) (Figure 3). The plasma n-6/n-3 ratio [ARA/(EPA+DHA)] was calculated to be 15% lower in the ATRA group compared to the control group (*p* = 0.015, two-tailed Student’s *t* test). Plasma levels of cholesterol, palmitic acid (16:0) and oleic acid (18:1) were similar in both experimental groups (Figure 3).

#### 2.1.4. Plasma Levels of Targeted Intermediate Metabolites

ATRA treatment did not significantly affect the plasma levels of major metabolites/cofactors in intermediary metabolism analyzed (hexose, hexose–phosphate, lactate, citrate–isocitrate, malate, AMP, nicotinamide, choline, and urea). However, trends toward decreased plasma levels of hexose-phosphate (46% decrease, *p* = 0.087, Student’s *t* test) and increased of lactate (~31-fold increase, p = 0.150, two-tailed Student´s *t* test) and malate (~7-fold increase, *p* = 0.087, two-tailed Student´s *t* test) were apparent in the ATRA-treated mice, compatible with changes in mitochondrial function and glucose metabolism (Figure 4). 

### 2.2. Gene Expression Analyses in Solid Tissues

Results of plasma analyses prompted us to assess effects of ATRA treatment on gene expression in selected tissues of key genes in carnitine, taurine and NO metabolism (Table 1). Expression of *Slc22a5*—encoding the plasma membrane carnitine transporter OCTN2—was higher in tissues of ATRA-treated mice compared to controls, especially in the liver. ATRA-treated mice also showed higher expression levels of *Aldh9a1*—encoding the rate limiting enzyme in carnitine biosynthesis, 4-trimethylaminobutyraldehyde dehydrogenase—in skeletal muscle and retroperitoneal WAT. In WAT and BAT depots, ATRA-treated mice had an increased expression of *Slc6a6*—encoding the taurine plasma membrane transporter protein TAUT—while expression of *Cdo1*—encoding cysteine dioxygenase 1 of the taurine biosynthetic pathway—was unaffected; gene expression of endothelial NO synthase (*Nos3*) was similarly unaffected. Expression levels of mitochondria oxidative metabolism- and thermogenesis (brown/beige adipocyte)-related genes in fat depots of animals of this study cohort have already been published and are up-regulated by ATRA treatment [8].

NMRI mice received a daily subcutaneous injection of ATRA at a dose of 50 mg/kg body weight (bw) or the vehicle (olive oil) during 4 days. Data are the mean ± SEM of 9–11 animals/group and are expressed relative to the mean value of the vehicle group, which was set to 100. Beta-actin (Actb) was used as a reference gene. rpWAT, retroperitoneal white adipose tissue; iWAT, inguinal white adipose tissue; BAT, brown adipose tissue. * Indicates a significant difference between the ATRA and the control group (*p* < 0.05 or *p* values indicated when 0.05 < *p* < 0.1, two-tailed Student´s *t* test).

### 2.3. Principal Component Analysis and Correlation Analyses

To help to identify the variables that better discriminate between control and ATRA-treated mice and the metabolic relationships between them, a PCA was conducted on a dataset of 117 variables comprising: relative plasma levels of all metabolites analyzed, energy efficiency, mass of fat depots, and mRNA levels of markers of oxidative/thermogenic capacity in BAT (interscapular) and WAT depots (inguinal, retroperitoneal and epididymal) of all animals in the experiment. 

The first three principal components (PC1, PC2, and PC3) explained over 50% (50.8%) of the total variance between the control and the ATRA group. Corresponding PCA biplots illustrated that PC2 was the most discriminating of the three (Figure 5). Loadings for the three components of all variables included in the analysis are shown in Appendix A. Considering the 15 variables with the higher loadings, PC1 was characterized by plasma levels of long- and medium-chain unsaturated ACs, and some AAs; PC2, by “classical” variables already known from previous studies to be affected by ATRA treatment—e.g., energy efficiency, fat depot mass, expression of *Ucp1* in BAT and of *Ppara* and *Pparg* in retroperitoneal WAT–together with certain plasma metabolites, namely carnitine, specific ACs (acetylcarnitine, AC C4OH, AC C8:0, and AC C16DC), DHA, taurine, and glutamine; and PC3, by a miscellaneous of variables. 

Pearson’s correlations between levels of plasma metabolites with the higher loadings in PC2 and variables related to BAT and WAT thermogenic capacity are shown in Appendix A. Plasma levels of acetylcarnitine (AC C2:0), AC C4OH, AC C16DC, and DHA significantly negatively correlated after Benjamini–Hochberg correction with values of energy efficiency, BAT mass and subcutaneous WAT mass, and positively with markers of thermogenic capacity in fat depots, namely *Ucp1* and *Ppara* mRNA levels in BAT, and *Ucp1* mRNA levels in visceral WAT. Similarly, plasma AC8:0 levels correlated negatively with energy efficiency and BAT mass, and positively with *Ucp1* mRNA levels in BAT. There were positive correlations between plasma carnitine, taurine, and glutamine levels with *Ucp1* mRNA levels in visceral WAT. 

### 2.4. PBMC Analyses

#### 2.4.1. Targeted Metabolomics in PBMC

The same molecules analyzed in plasma were analyzed in PBMC. The results indicated a generalized trend toward lower levels of almost all molecules identified, except AC C14:1 and EPA, in PBMC of ATRA-treated mice compared to controls (Figure 6A–D). This trend to reduction reached statistical significance (*p* < 0.05, Mann–Whitney *U* test) for AC C5:0, several AAs (aspartate, glutamine, histidine, ornithine, serine, and taurine), and oleic acid, according to Mann–Whitney *U* test (*p* < 0.05), but significances disappeared after correction for multiple tests using the Benjamini–Hochberg procedure.

#### 2.4.2. Gene Expression in PBMC

PBMC express retinoid receptors transcription factors (RAR and retinoid X receptor) [37], and we previously showed that gene expression in whole blood partly reflects transcriptional changes induced by ATRA treatment in internal tissues [38]. Here, we analyzed mRNA expression levels of target genes, related to ATRA metabolism and lipid and energy metabolism, in isolated PBMC. As shown in Figure 6E, ATRA treatment significantly up-regulated *Cpt1a* (~7-fold increase) and *Cyp26a1* (~130-fold increase) and down-regulated *Fasn* (by 75%) expression in PBMC. A down-regulatory effect on *Srbp1c* was also evidenced (by 80%, *p* = 0.086, Mann–Whitney *U* test). *Cpt1a* encodes the liver isoform of the rate limiting enzyme for mitochondrial FAO (carnitine palmitoyltransferase 1); as in PBMC, expression of *Cpt1* is induced in adipose, liver, and skeletal muscle tissues and cells following ATRA treatment [6,7,9,10,11,12,13,15]. *Cyp26a1* is a classic RAR target gene induced under conditions of abundant ATRA and encoding an enzyme of ATRA catabolism [39]; its massive induction in PBMC strongly suggests that, following its subcutaneous injection, ATRA effectively spread through the body and entered the blood cells. *Fasn* (encoding fatty acid synthase) and *Srbp1c* (encoding sterol regulatory element binding protein) are lipogenesis-related genes and their down-regulation was previously reported in the liver of ATRA-treated mice [12]. Expression in PBMC of other genes analyzed—namely: *Ucp1* and *Ucp2*, encoding thermogenesis related proteins; *Acox1*, encoding the rate limiting enzyme of peroxisomal FAO; and *Aldh9a1*, related to carnitine biosynthesis—was not affected by ATRA treatment.

## 3. Discussion

Experimental studies in animals [40,41] and a prospective observational study in humans [42] point to beneficial effects of ATRA on adiposity and metabolic health. We here identified through targeted metabolomics changes in circulating markers in response to a short-term, high-dose ATRA treatment in mice that may give further clues about the mechanisms of ATRA action on metabolic health and energy and lipid metabolism in mammalian tissues, notably the adipose tissues. Main findings and their implications are discussed next.

### 3.1. Plasma Markers of Enhanced FAO in Tissues of ATRA-Treated Mice 

Generalized higher levels of AC species in plasma of ATRA-treated mice relative to controls is in overall concordance with ATRA favoring the mobilization of fat stores and FAO in tissues (reviewed in [40,41]). Higher levels of C2:0 AC (acetylcarnitine) is in keeping with increased oxidation of FAs, and possibly other fuel substrates, to acetyl-CoA. The AC C4OH peak found to be increased in the ATRA mice may represent various stereoisomers, namely: D-C4OH-carnitine, derived from the CoA ester of the ketone body D-3-hydroxybutyrate; L-C4OH carnitine, derived from L-3-hydroxybutyryl-CoA, a FAO intermediate; or L-isoC4OH-carnitine, derived from L-3-hydroxyisobutyryl-CoA, an intermediate in valine catabolism [24]. We favor the first interpretation because ATRA treatment increases ketone body production and blood levels [12], while it had no effect on the circulating levels of valine (Figure 2) or AC C3:0 and AC C5:0 (Figure 1), which are ACs produced during the catabolism of valine and other branched-chain AAs. Increased plasma levels of AC C8:0 may reflect increased peroxisomal long-chain FAO, which produces medium-chain ACs that are released from the peroxisome to eventually complete their oxidation in mitochondria [43]. Whereas increased plasma levels of AC C16DC may reflect increased endoplasmic reticulum omega-oxidation of long-chain FAs, a pathway active mainly in the liver that renders dicarboxylic FAs for peroxisomal beta-oxidation [44,45]. Finally, the trend to increased plasma levels of ACs of the C18:1-C16:1-C14:1-C12:1 series in ATRA-treated mice is suggestive of increased mobilization and oxidation of oleic acid (18:1), one of the most abundant FA in human and also murine WAT [46]. 

The tissue origin of excess plasma ACs in ATRA mice may be multiple. We favor a hepatic origin for a number of reasons. First, recent studies indicate liver has a central role in whole body AC metabolism and is a major contributor to systemic ACs [47,48]. Second, exposure to ATRA increases the conversion of exogenous palmitate to both CO_2_ and acid soluble products including ACs in cells of hepatic origin (HepG2) [13], while in skeletal muscle cells (C2C12 myotubes) it increases palmitate oxidation to CO_2_ but decreases palmitate oxidation to acid soluble products [11]. Third, compared to control mice, ATRA-treated mice had significantly increased gene expression of the plasma membrane carnitine transporter OCTN2 selectively in the liver (Table 1). ATRA treatment was previously demonstrated to enhance the expression or activity in liver of other proteins involved in AC metabolism, such as CPT1a, mitochondrial carnitine/acylcarnitine translocase, and carnitine acetyltransferase [12,13,15,49]. 

Increased carnitine (and acetylcarnitine) levels in plasma of ATRA-treated mice may reflect a more effective re-entering of circulating ACs into FAO in extra-hepatic tissues in these mice than in controls. Enhanced carnitine biosynthesis cannot be discarded, and is in fact suggested by increased *Aldh9a1* expression in skeletal muscle and retroperitoneal WAT of ATRA mice (Table 1). ATRA-dependent induction of *Aldh9a1* in WAT is in agreement with a previous report [10]. Etiologically, it seems logical that conditions/agents that promote lipolysis and FAO favor simultaneously an increase in the availability of the carnitine necessary for this metabolism; this is so for fasting [50] and appears to be the case for ATRA (this work).

The pathophysiological significance of ACs is complex, since, as stated in the Introduction, increased levels in plasma and tissues are observed in obesity and insulin resistant state [25,26,27,28,29], but also after weight loss and exercise [30,31,32]. It appears that, depending on the conditions, excess ACs may inflict insulin resistance or simply reflect FAO rates in tissues [24]. Further, AC synthesis and efflux from mitochondria serves physiological functions in the modulation of cellular metabolism, as it helps avoiding both the sequestration of mitochondrial CoA in the form of acyl-CoAs and the feed-back inhibition of the pyruvate dehydrogenase complex (and hence of glucose oxidation) by excess mitochondrial acetyl-CoA [24,51]. Interestingly, increments in long-chain AC levels found under conditions of obesity, high-fat diet, or lipid stress take place in parallel with decreased tissue and plasma levels of free carnitine [52,53], whereas increments of AC levels found under conditions of weight loss or exercise take place in parallel with increases in the tissue/ plasma levels of free carnitine [30,32]. In fact, carnitine insufficiency is a common feature of insulin-resistant states such as advanced age and diet-induced obesity in rodents [54]. In this context, the finding of higher plasma levels of both free carnitine and ACs accompanying body weight and fat loss and increased glucose tolerance and insulin sensitivity in ATRA mice is remarkable (this work and [6,21,22]). Our results support the concept that, under conditions of high FAO rate and availability of carnitine, synthesis and secretion of ACs, with subsequent elevation of plasma levels, can serve to favor mitochondrial activity and cellular glucose metabolism. To be noted, dietary supplementation with carnitine or acetylcarnitine potentiates mitochondrial function and metabolic control in rodents [54,55] and humans with type 2 diabetes [56]. Overall, it is suggested that increments in plasma levels of carnitine and acetylcarnitine brought about by ATRA treatment may contribute benefits to metabolic health.

The tendency of ATRA mice to present higher levels of plasma creatine than controls may be secondary to ATRA promotion of FAO especially in skeletal muscle [9,10,11]. Thus, under conditions of enhanced fat utilization, muscle energy metabolism would rest less in the creatine–phosphocreatine system, so that less creatine would be taken from the circulation, leading to increased levels in plasma (skeletal muscle has virtually no creatine-synthesizing capacity [57]). In favor of this interpretation, the entry of [^14^C-creatine] (after intraperitoneal injection in mice) into muscles is inhibited under fasting conditions [58], when muscle FAO is increased, whereas depletion of intramuscular creatine increases glucose and FA uptake and FAO in skeletal muscle [59], suggesting the existence of a compensatory regulatory mechanism. Other explanations cannot be discarded, including enhanced creatine synthesis in competent tissues in the ATRA mice.

Higher plasma lactate (trend) in ATRA-treated mice could reflect decreased lactate uptake by tissues under increased use of FAs as a fuel promoted by ATRA. Combined changes in plasma levels of lactate (increased) and hexose-phosphate (decreased) in ATRA mice relative to controls may also be consistent with greater glucose utilization by the glycolytic pathway. Finally, higher plasma malate levels (trend) in the ATRA mice may reflect an enhanced capacity for cellular oxidative metabolism through the Krebs cycle [60]. Interestingly, increased plasma malate was highlighted as a trait reflective of improved metabolic health after a weight loss and exercise intervention in sedentary, obese insulin-resistant women [60].

### 3.2. Plasma Markers of ATRA Treatment that May be Linked to Enhanced Oxidative Metabolism and Thermogenesis in Adipose Tissues

ATRA is a brown fat activator and an agent inducing the acquisition of BAT-like properties in WAT (so-called WAT browning) in rodents through both direct effects in adipocytes and systemic effects [3,4,5,6,7,8,11]. A number of plasma markers identified in this work, summarized in Figure 7, are suggestive of novel indirect mechanisms that could contribute to effects of ATRA treatment on energy metabolism in adipose tissues. Among them are increased plasma ACs and creatine levels already discussed above. Secreted ACs are a transport form of FA, and it has recently been shown that, under cold exposure, AC production by the liver provides ACs to be used in BAT as fuels for UCP1-dependent thermogenesis [61]. Interestingly, we found significant positive correlations between *Ucp1* mRNA expression levels in BAT and plasma levels of various AC species (Appendix A), thus supporting the notion of co-regulation of BAT thermogenesis and AC production (likely) in the liver, and the extension of this integrated response to pharmacological stimulus. As for increased plasma creatine levels in ATRA mice, they may reflect increased production and local effects in adipose tissues, where synthesis and metabolism of creatine plays a role in supporting beige and brown fat UCP1-independent thermogenesis induced by cold or diet [62,63]. Other markers to be considered here and discussed next are increased GABR, plasma taurine, and glycine and plasma n-3 PUFA in the ATRA-treated mice (Figure 7).

The GABR result suggests ATRA can increase NO synthesis, which in fact is demonstrated in ATRA-exposed endothelial cells in culture [64,65]. NO enhances mitochondriogenesis and BAT thermogenic activity [66,67], and the NO second messenger cGMP promotes WAT browning [68]. Beyond effects on adipose tissues, NO is critical for vascular health [69], and its decreased bioavailability is a main underlying factor of endothelial dysfunction in the metabolic syndrome (MetS) [70]. Stimulatory effects of ATRA on NO production may help explaining epidemiological findings pointing to circulating ATRA levels as a predictor of MetS in humans independently of adiposity and insulin resistance [42]. To be noted, GABR is decreased in humans at high risk of diabetes and cardiovascular disease [35,36] and animal models of MetS [71,72], and it normalizes following therapeutic intervention in patients with type 2 diabetes [73]. A metabolomics study in mice identified increased plasma citrulline, leading to decreased GABR, as a mark of diet-induced obesity predictive of MetS development [71]. Lower plasma citrulline and increased GABR found here in the ATRA mice group are, therefore, in keeping with overall beneficial effects on metabolic health.

Together with glycine, plasma taurine is found to be reduced in obese and diabetic subjects and animal obesity models [74,75], and dietary supplementation of these AAs ameliorates diet-induced obesity and metabolic sequelae in animal models and, according to some studies, humans [74,75,76]. Anti-obesity effects of taurine have been ascribed to increases in resting energy expenditure linked to an enhancement of mitochondrial oxidative metabolism in adipose tissues [77,78]. Taurine and glycine are specifically used for bile acid conjugation, and their higher plasma levels (especially of taurine) in the ATRA group could reflect ATRA-induced inhibition of hepatic bile acid synthesis, which has been described in mice fed an ATRA-supplemented diet [79]. Increased renal reabsorption is envisaged as another contributing mechanism, since ATRA stimulates TAUT gene (*Slc6a6/Taut*) expression and taurine uptake in a renal epithelial human cell line [80]. We here addressed ATRA-induced changes in the capacities for taurine biosynthesis and secretion by adipose tissues; while biosynthetic *Cdo1* mRNA levels were unaffected, increased expression of *Slc6a6/Taut* found in all analyzed fat depots in ATRA mice suggests ATRA may enhance the efficiency of taurine export from adipose tissues. This is of interest considering that WAT-born taurine has been proposed to function as a sort of adipokine with anti-obesity effects and repressed in obesity [77].

n-3 PUFA have been related to multiple beneficial actions on metabolism which include the activation of thermogenesis in BAT and browning of WAT [81,82]. A recent meta-analysis provided substantial evidence of a higher circulating n-3 PUFA (especially DHA and docosapentaenoic acid) associated with a lower MetS risk in humans [83]. Higher plasma levels of n-3 PUFA (especially DHA) found in ATRA-treated mice relative to controls are, therefore, of interest. Decreased plasma n-6/n-3 ratio in the ATRA group is also of interest, since n-6 and n-3 PUFA have opposite effects on systemic inflammation, adipogenesis and WAT browning, and high n-6/n-3 ratios in plasma have been related to increased risk for obesity [84,85]. Higher plasma n-3 PUFA levels in the ATRA mice may conceivably reflect increased n-3 PUFA synthesis in the liver and in metabolically activated adipose tissues. ATRA treatment in mice induces in liver a collection of genes in the biosynthesis of unsaturated FAs responsible for anti-inflammation [14]. Activated BAT and browned WAT could be additional sources of DHA as suggested by previous works showing that (i) tissues with high oxidative metabolism tend to have high DHA levels, (ii) long-chain PUFA synthetic desaturase enzymes reside in mitochondria, and (iii) differentiated brown but not white adipocytes in culture synthesize DHA, supporting altogether the hypothesis that functional BAT is a net producer of DHA [86].

### 3.3. PBMC Markers of ATRA Treatment

ATRA treatment appeared to affect in opposite directions the relative levels of AC species in PBMC (down-regulation) and plasma (up-regulation). Considering that the efflux of ACs from tissues/blood cells to plasma is concentration-dependent [87], this suggests little contribution of PBMC to the plasma AC pool under our experimental conditions. Down-regulation of the relative levels of most studied metabolites in PBMC of ATRA-treated mice may reflect distinct specific effects of ATRA on these cells. In fact, it is known that ATRA modulates inflammatory responses and the differentiation of immune cells [88] and inhibits ATP-consuming cellular activities in PBMC [89]. Nevertheless, gene expression changes in PBMC, i.e., the up-regulation of *Cpt1a* and *Cyp26a1* and the down-regulation of lipogenic *Fasn* and *Srebp1c* genes, are reminiscent of transcriptional effects in solid tissues elicited by ATRA treatment in vivo [6,7,9,10,11,12,13,15]. Thus, PBMC provided mRNA transcript-based markers of systemic effects of ATRA treatment, whereas targeted metabolomics analyses results most likely reflected PBMC-specific ATRA effects. We already described the induction of *Cpt1a* and *Cyp26a1* in whole blood of ATRA-treated mice [38]. Expression of *Cpt1a* in PBMC is particularly sensitive to changes in the animal’s nutritional and energy status [90].

### 3.4. Summary Discussion

In summary, this work identified through targeted metabolomics circulating markers of the interaction of a treatment with a high dose of ATRA, the main carboxylic form of vitamin A, with oxidative metabolism and metabolic health-related pathways. Novel markers revealed by plasma analyses included increased levels of carnitine, acetylcarnitine and certain longer AC species; decreased levels of citrulline and increased GABR for NO synthesis; increased levels of taurine + glycine, and possibly creatine; increased levels of DHA; and a decreased n-6/n-3 PUFA ratio. Some of these markers (such as increased ACs) most likely reflect the stimulation of lipid mobilization and oxidation promoted by ATRA systemically and particularly in the liver. Other markers identified—such as increased plasma carnitine, creatine, taurine + glycine, and DHA, lower plasma citrulline, or higher GABR—may play a more active or causal role underlying ATRA effects, since previous studies indicate that their increased availability can improve mitochondrial function and metabolism and favorably impact other physiological functions of relevance for metabolic and cardiovascular health. Results in this work connect ATRA to numerous nutrition-modulated biochemical pathways, and suggest novel potential mechanisms of action of local vitamin A-derived retinoic acid on metabolic health.

## 4. Materials and Methods

### 4.1. Animal Study

The animal experiment was conducted following international standards for the use and care of laboratory animals and approved by the Bioethical Committee of the University of the Balearic Islands (Ref. 3513, 26/03/12). Briefly, twelve-week-old NMRI male mice (CRIFFA, Barcelona, Spain) fed ad libitum regular laboratory chow (Panlab, Barcelona, Spain; 2.25 IU vitamin A/kcal; 3339 kcal/kg) received one daily subcutaneous injection of ATRA at a dose of 50 mg/kg body weight during the 4 days before they were sacrificed, a widely experienced dose and protocol [6,8,9,16,21]; control mice were injected the vehicle (100 μL olive oil) (*n* = 11 animals/group). The animals were housed 2–3 per cage, at 22 °C under 12 h light/12 h dark cycles (lights on at 08:00). Body weight and food intake (the latter on a per-cage basis) were followed daily during the treatment period. Right prior euthanization, blood from the facial vein was collected for PBMC isolation (200–400 μL of blood in 100 μL of 3.8% sodium citrate) and for plasma preparation (~50 μL blood in a heparinized capillary tube). Plasma was obtained by centrifugation (1000× *g* for 10 min at 4 °C) and stored at −80 °C until further analysis. The animals were euthanized by decapitation at the start of the light cycle. Tissues including the liver, epididymal, inguinal, and retroperitoneal WAT, and interscapular BAT were excised in their entirety, weighted, snap-frozen in liquid nitrogen, and stored at −80 °C. The sum of the weight of the individual WAT depots as percentage of body weight was used as adiposity index. Energy efficiency during the treatment period was defined as body weight change divided by the amount of calories eaten.

### 4.2. Isolation of PBMC

A flotation method was used according to an on-line published protocol (from Alere Technologies AS, Oslo, Norway; https://www.axis-shield-density-gradient-media.com/C07.pdf; last accessed 15 April 2019). Briefly, blood in sodium citrate was transferred to a 15 mL tube, and 5 mL of freshly prepared OptiPrep™ (Sigma-Aldrich, St. Louis, MO, USA) working solution (1.5 mL of OptiPrep™ in 5 mL of Tricine-buffered saline (TBS), pH 7.4) were added, mixing gently by inversion. The mixture was carefully covered with 0.5 mL TBS and centrifuged at 1000× *g* for 30 min at 20 °C, after which the PBMC were collected from the meniscus downwards to half a centimeter of the erythrocyte pellet. The cell suspension was diluted in two volumes of TBS and centrifuged at 350× *g* for 10 min at 20 °C. Finally, the cell pellet was suspended in 120 μL of phosphate buffered saline, pH 7.2 and stored in two aliquots at −80 °C until further analysis. The number of viable cells was estimated by direct counting in a Neubauer chamber using Trypan blue exclusion dye (Sigma-Aldrich). PBMC were successfully isolated from 3 control and 5 ATRA mice, yielding 5.8 × 10^5^ ± 1.2 × 10^5^ viable cells from 200–400 μL of blood.

### 4.3. Targeted Metabolomics of Plasma and PBMC

Levels of ACs and AAs were simultaneously semiquantified, together with specific FA species and key metabolic intermediates, in 3 µL of plasma or PBMC suspension (~14,000 cells) by using liquid chromatography-tandem mass spectrometry (LC-MS/MS) based on a previously used method [91,92,93]. The analysis was performed on UltiMate 3000 Rapid Separation system (RSLC) (Thermo Fisher Scientific, Waltham, MA, USA) coupled to the 5500 QTRAP mass spectrometer (AB Sciex LLC, Framingham, MA, USA). Briefly, samples were mixed with internal standards in 100 µL of acetonitrile with 0.1% acetic acid, vortexed, precipitated proteins removed by centrifugation at 12,000 g, 4 °C, 10 min, and immediately analyzed. The internal deuterated standards for ACs and AAs (MassChrom Newborn Screening Kit, Chromsystems, Gräfelfing, Germany) were used. Samples (2 L) were separated using Luna NH2 HPLC column (150 × 2.1 mm Phenomenex), gradient elution: A, acetonitrile; B, 20 mM ammonium acetate, pH 9.45, flow 0.3 mL/min; 0 min–5%B, 3.8 min–70%B, 4.3 min–95%B, 8 min–95%B, 9 min–5%B, 12 min–5%B. Analytes were detected in multiple reaction monitoring (MRM) positive/negative ion-switching mode as before [92]. Data analysis was performed with Analyst software (AB Sciex LLC, Framingham, MA, USA). Peak area raw data were normalized by the plasma volume or PBMC number.

### 4.4. Total RNA Isolation and Real Time Quantitative PCR (RT-qPCR)

Total RNA from PBMC (~5 × 10^5^ cells in 100 μL) was isolated using Direct-zol^TM^ RNA MiniPrep (Zymo Research, Irvine, CA, USA), and purified with E.Z.N.A. MicroElute RNA Clean Up (Omega Bio-Tek, Norcross, GA, USA). Total RNA from tissues was isolated using TRIzol reagent (Thermo Fisher Scientific, Waltham, MA, United States). RNA was quantified using NanoDrop ND-1000 spectrophotometer (NanoDrop Technologies Inc., Wilmington, DE, USA) and its integrity and purity confirmed by 1% agarose gel electrophoresis. Total RNA from PBMC (0.05 μg) and tissues (0.25 μg) was reverse transcribed using iScript™ cDNA synthesis kit (Bio-Rad Laboratories, Madrid, Spain) and a MuLV Reverse Transcriptase-based dedicated kit (Life Technologies, Grand Island, NY, USA), respectively, following the manufacturers’ instructions. Real-time PCR was used to semi-quantify mRNA expression levels of genes of interest from the cDNA. Reverse transcription and amplification reactions were carried out in an Applied Biosystems 2720 Thermal Cycler (Applied Biosystems, Madrid, Spain), using StepOnePlus^TM^ Real-Time PCR System (Applied Biosystems). Each PCR was performed from diluted cDNA template, forward and reverse primers (1 mM each), and Power SYBR Green PCR Master Mix. Primers were obtained from Sigma (Madrid, Spain); sequences are available on request. In order to verify the purity of the products, a melting curve was produced after each run according to the manufacturer’s instructions. The threshold cycle (Ct) was calculated by the instrument’s software (StepOne Software v2.0), and the relative expression of each mRNA was calculated using the Pfaffl method [94]. Beta-actin (Actb) was used as a reference genes.

### 4.5. Statistical Analyses

All data are expressed as the mean ± SEM. Comparisons between the control and the ATRA group were assessed by Student’s *t* test or nonparametric Mann–Whitney *U* test. Unsupervised principal component analysis (PCA) was carried out with log-transformed data. Principal components were considered as significant if they contributed more than 6% to the total variance. Correlation between plasma metabolites and gene expression and biometric parameters was analyzed by Spearman’s correlation test. Threshold of significance was set at *p* < 0.05. The Benjamini–Hochberg false discovery rate (FDR) procedure was used on metabolite levels (63 molecular species) and correlations (80 correlations) to control for multiple testing, with a significance level of *p* < 0.16. IBM SPSS Statistics for Windows, version 19.0 (IBM Corp., Armonk, NY, USA) was used for the analyses.

## Figures and Tables

**Figure 1 ijms-20-03640-f001:**
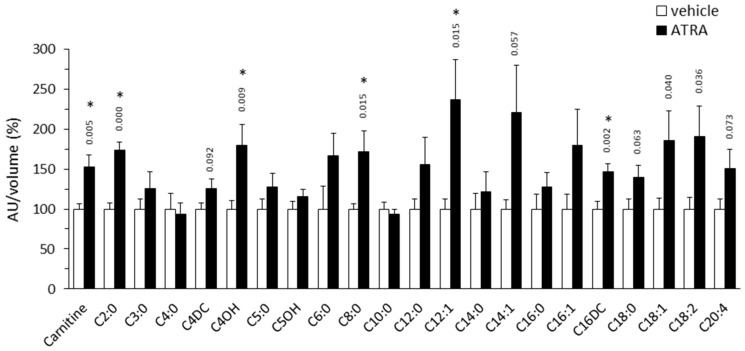
Relative plasma levels of carnitine and the indicated acylcarnitines species in NMRI mice treated with all-trans retinoic acid (ATRA; 50 mg/kg body weight, subcutaneous injection) or the vehicle (olive oil) during 4 days. Metabolite peak area (as arbitrary units, AU) was normalized by the volume of plasma used in LC-MS/MS. Data are expressed as percentage with respect to the normalized value in the control group, which was set to 100%, and are the mean ± SEM of 11 animals per group. * Indicates significant difference between the ATRA and the control group after Benjamini–Hochberg correction (observed *p* values in two-tailed Student’s *t* test are indicated when *p* < 0.1).

**Figure 2 ijms-20-03640-f002:**
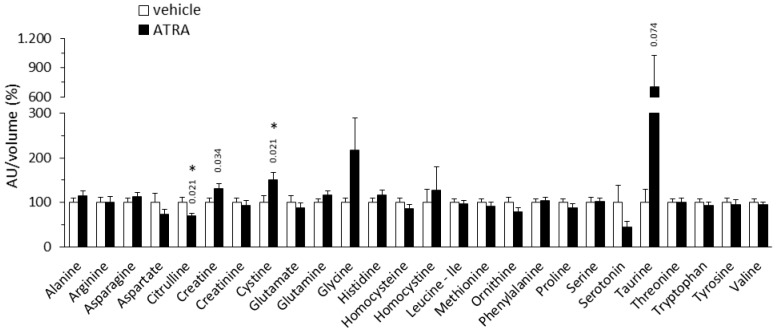
Relative plasma levels of amino acids in NMRI mice treated with all-trans retinoic acid (all-trans retinoic acid (ATRA; 50 mg/kg body weight, subcutaneous injection) or the vehicle (olive oil) during 4 days. Metabolite peak area (as arbitrary units, AU) was normalized by the volume of plasma used in LC-MS/MS. Data are expressed as percentage with respect to the normalized value in the control group, which was set to 100%, and are the mean ± SEM of 11 animals per group. * Indicates significant difference between the ATRA and the control group after Benjamini–Hochberg correction (observed *p* values in two-tailed Student’s *t* test are indicated when *p* < 0.1).

**Figure 3 ijms-20-03640-f003:**
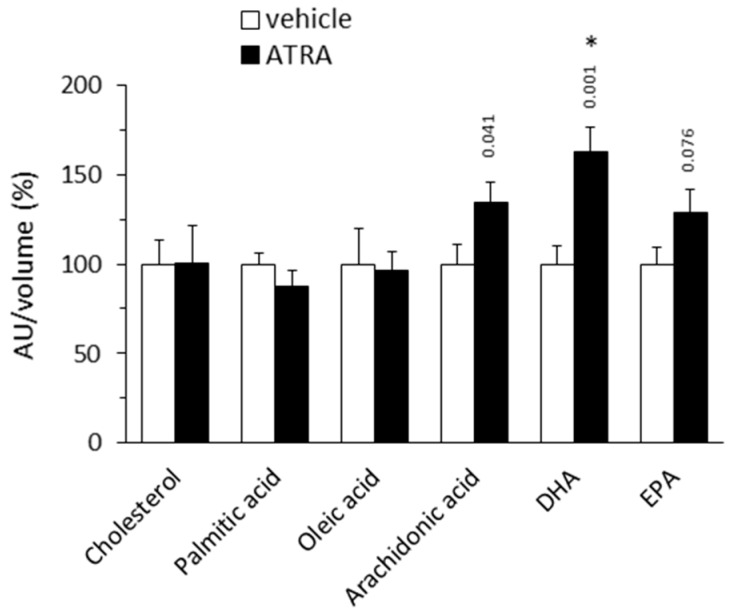
Relative plasma levels of the indicated lipids in NMRI mice treated with all-trans retinoic acid (ATRA; 50 mg/kg body weight, subcutaneous injection) or the vehicle (olive oil) during 4 days. Metabolite peak area (as arbitrary units, AU) was normalized by the volume of (plasma used in LC-MS/MS. Data are expressed as percentage with respect to the normalized value in the control group, which was set to 100%, and are the mean ± SEM of 11 animals per group. * Indicates significant difference between the ATRA and the control group after Benjamini–Hochberg correction (observed *p* values in two-tailed Student’s *t* test are indicated when *p* < 0.1). DHA, docosahexaenoic acid; EPA, eicosapentaenoic acid.

**Figure 4 ijms-20-03640-f004:**
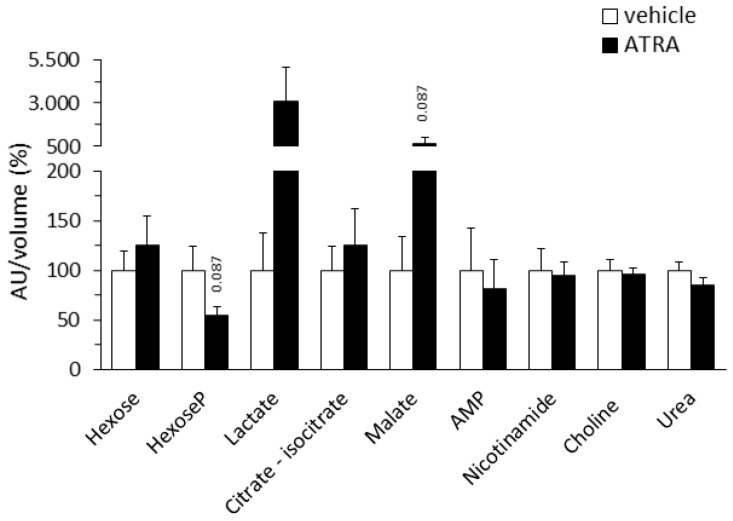
Relative plasma levels of the indicated compounds of intermediary metabolism in NMRI mice treated with all-trans retinoic acid (ATRA; 50 mg/kg body weight, subcutaneous injection) or the vehicle (olive oil) during 4 days. Metabolite peak area (as arbitrary units, AU) was normalized by the volume of plasma used in LC-MS/MS. Data are expressed as percentage with respect to the normalized value in the control group, which was set to 100%, and are the mean ± SEM of 11 animals per group. * Indicates significant difference between the ATRA and the control group after Benjamini–Hochberg correction (observed *p* values in two-tailed Student’s *t* test are indicated when *p* < 0.1).

**Figure 5 ijms-20-03640-f005:**
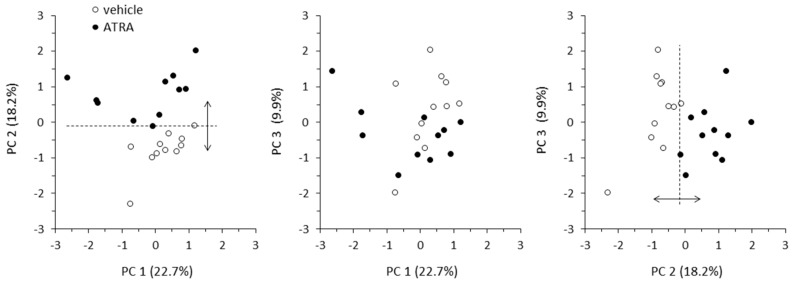
Score biplot representation of the first three principal components (PCs) in Principal Component Analysis. Data (of 11 animals per group) were spread by treatment to assess possible relationships between 117 variables (shown in Appendix A).

**Figure 6 ijms-20-03640-f006:**
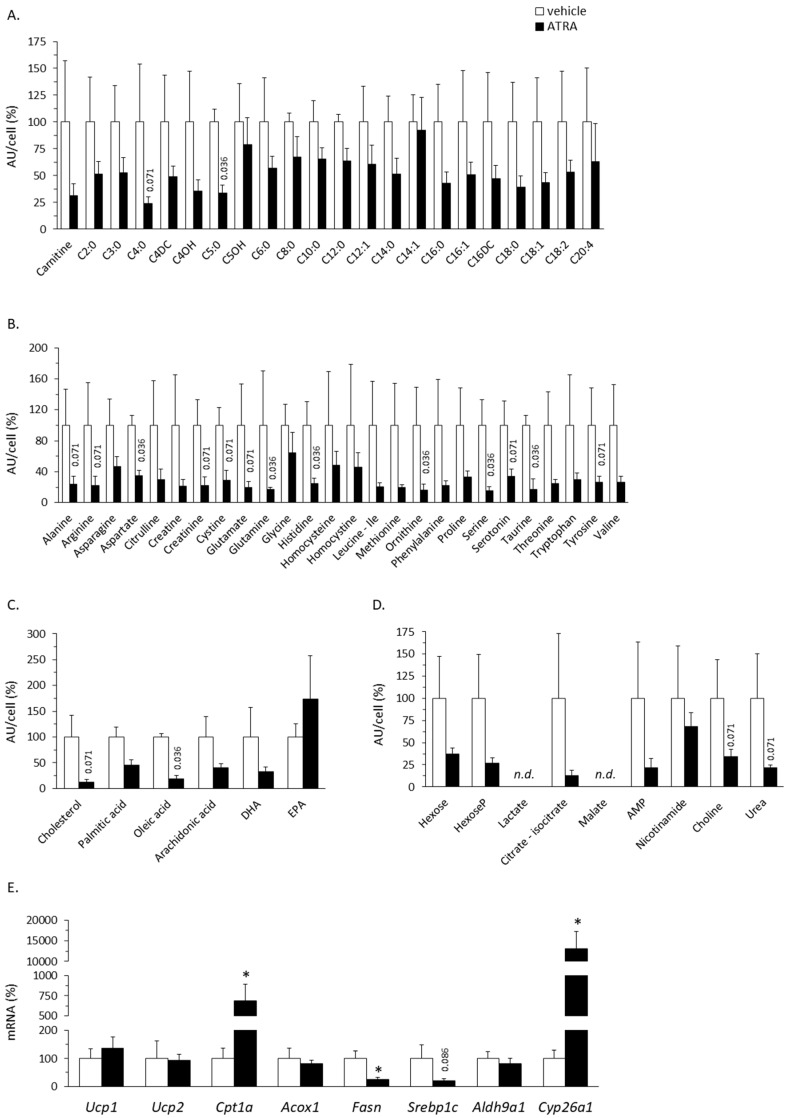
Relative plasma levels of carnitine and acylcarnitines (**A**), amino acids (**B**), the indicated fatty acids (**C**), the indicated metabolites/cofactors of intermediary metabolism (**D**), and the indicated mRNAs (**E**) in PBMC of mice treated with all-trans retinoic acid (ATRA; 50 mg/kg body weight, subcutaneous injection) or the vehicle (olive oil) during 4 days. Data are the means ± SEM of 3–5 mice per group and are expressed relative to the mean value of the control (vehicle-treated) group, which was set to 100. In **A**–**D**, metabolite peak area (as arbitrary units, AU) was normalized per cell, and observed *p* values in Mann–Whitney *U test* are indicated when *p* < 0.1 (significances disappeared after Benjamini–Hochberg correction). In **E**, mRNA levels were normalized to the mRNA levels of the reference gene Actb, * indicates a significant difference between the ATRA and the control group (*p* < 0.05), and *p* values are indicated when 0.05 < *p* < 0.1 (Mann–Whitney *U* test).

**Figure 7 ijms-20-03640-f007:**
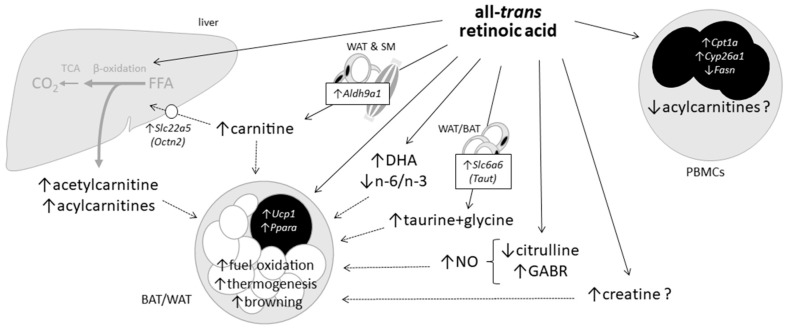
Effects of short-term, high-dose all-trans retinoic acid (ATRA) treatment in mice revealed in this work and suggested connection to oxidative metabolism in adipose tissues. Direct or indirect ATRA effects on metabolite levels and gene expression are indicated by arrows. Question marks are used to distinguish observed trends from statistically significant effects.

**Table 1 ijms-20-03640-t001:** mRNA levels of targeted genes in selected tissues of control and all-trans retinoic acid (ATRA)-treated mice.

	Vehicle (Olive Oil)			ATRA (50 mg/kg bw)		
*Slc22a5 (Octn2)*						
liver	100	±	27	203	±	34 *
skeletal muscle	100	±	13	133	±	26
rpWAT	100	±	24	184	±	58
iWAT	100	±	19	143	±	40
*Aldh9a1*						
liver	100	±	11	109	±	13
skeletal muscle	100	±	11	156	±	20 *
rpWAT	100	±	16	249	±	32 *
iWAT	100	±	14	124	±	27
*Slc6a6 (Taut)*						
rpWAT	100	±	10	258	±	35 *
iWAT	100	±	17	160	±	27 ^0.068^
BAT	100	±	11	137	±	11 *
*Cdo1*						
rpWAT	100	±	10	80	±	8
iWAT	100	±	16	70	±	17
BAT	100	±	9	116	±	12
*Nos3 (eNos)*						
rpWAT	100	±	16	95	±	8
iWAT	100	±	18	100	±	25
BAT	100	±	11	114	±	16

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
