# Peer review of "Novel Markers of the Metabolic Impact of Exogenous Retinoic Acid with A Focus on Acylcarnitines and Amino Acids"

_ijms, 2019, doi:10.3390/ijms20153640_

Round 1

Reviewer 1 Report

First of all, the authors provided novel mechanisms of action of ATRA on metabolic health by linking ATRA to nutrition-modulated pathways. 

It is well written and read — the experiments designed meticulously. The references are up-to-date. The discussion captured the data presented, and it was also provocative. The introduction should provide a bit more background. 

One is using some of the figures in the main manuscript rather than in the supplementary. I recommended placing supplementary figure two and three as regular figures. 

I would also suggest including PPARgamma and RARbeta as target genes related to ATRA metabolism and lipid and energy metabolism in isolated PBMCs. 

Author Response

We appreciate the constructive criticisms and suggestions raised by the Reviewers and thank them very much for these comments, as we believe the manuscript has gained by taking them into account.

Reviewer 1

First of all, the authors provided novel mechanisms of action of ATRA on metabolic health by linking ATRA to nutrition-modulated pathways.  It is well written and read — the experiments designed meticulously. The references are up-to-date. The discussion captured the data presented, and it was also provocative.

We are happy by these comments, thank you.

The introduction should provide a bit more background.

Information regarding the complex interpretation of increases in acylcarnitine levels has been moved from the Discussion to the Introduction (last paragraph), where we agree it is better placed.

One is using some of the figures in the main manuscript rather than in the supplementary. I recommended placing supplementary figure two and three as regular figures.

Following this Reviewer’s comment, former supplementary figures two and three have been joined in a single regular figure (Figure 6).

I would also suggest including PPARgamma and RARbeta as target genes related to ATRA metabolism and lipid and energy metabolism in isolated PBMCs.

RARbeta, but not alpha or gamma isoforms, is indeed selectively induced in white adipose tissue depots of ATRA-treated mice (Berry and Noy 2008 PubMed PMID 19364826, and our unpublished results), but not in the PBMC (relative expression levels, control group: 100±39.5; ATRA group: 68±19, n=3-5 animals per group, our unpublished results). These results are envisaged to be part of a different manuscript, as in the present one gene expression analyses were focused mainly on potential downstream target genes coding for metabolic enzymes and uncoupling proteins, rather than on transcription factors mediating ATRA effects.  For PPARgamma, we previously showed that its mRNA levels in whole blood are unaffected by ATRA treatment under the same conditions as used here (Petrov et al. 2016 PubMed PMID 27163124).

Reviewer 2 Report

Ribot et al perform a targeted metabolomics analysis of mice treated with atRA, a compound known to stimulate mitochondrial uncoupling via UCP1 in BAT and fat oxidation in WAT, lowering body weight. The paper is very well written, the experimental design is sound, and the data are clearly presented. But more information is needed on the analytical methods regarding the metabolomics analysis before further consideration and the statistical analyses need to be redone. Overall, the results are preliminary and the conclusions speculative, though interesting.

- p-values from the metabolomics data should be corrected for multiple testing to avoid false positives, see Benjamini-Hochberg procedure. The same approach should be taken for the correlation data. This is a major revision, as many of the significant changes highlighted and discussed, may no longer be significant.

- More information is needed regarding the FIA analysis, such as instrument conditions and sample preparation. The reference is insufficient.

- Does the kit include internal standards? Could data be presented as absolute concentrations instead of arbitrary units?

- Without internal standards to control for differences in ionization efficiency between different metabolites and to control for differences in ion suppression between different samples, summing metabolite responses can be misleading. How is this dealt with here?

- It is well known that ATP can fragment into ADP and AMP in the MS source during ionization, and if not separated chromatographically, biological AMP can not be distinguished from the AMP signal that arises from ATP fragmentation in the source. How is this dealt with here?

- Metabolomics analysis of tissues involved in energy homeostasis (WAT, BAT, and liver) may be informative and provide answers to questions raised in the discussion.

Author Response

We appreciate the constructive criticisms and suggestions raised by the Reviewers and thank them very much for these comments, as we believe the manuscript has gained by taking them into account.

Reviewer 2

Ribot et al perform a targeted metabolomics analysis of mice treated with atRA, a compound known to stimulate mitochondrial uncoupling via UCP1 in BAT and fat oxidation in WAT, lowering body weight. The paper is very well written, the experimental design is sound, and the data are clearly presented.

We are happy by these comments, thank you.

But more information is needed on the analytical methods regarding the metabolomics analysis before further consideration and the statistical analyses need to be redone. Overall, the results are preliminary and the conclusions speculative, though interesting.

See our answers below to the specific points in which the Reviewer broke down his/her general point here. Still, the conclusions are to some degree speculative, because the type of data in the manuscript (basically targeted metabolomics and gene expression analyses) are compatible with the involvement of a variety of metabolic pathways in mediating effects of exogenous ATRA, but not definitive probe of this involvement (which would require the use of many other models).
We are happy the Reviewer considered the results interesting.

- p-values from the metabolomics data should be corrected for multiple testing to avoid false positives, see Benjamini-Hochberg procedure. The same approach should be taken for the correlation data. This is a major revision, as many of the significant changes highlighted and discussed, may no longer be significant.

We have now applied the Benjamini-Hochberg procedure for the control of false positive results in multiple testing, to both the targeted metabolomics data (63 molecular species) and the correlation data (80 correlations), as requested. This is now indicated in the statistical analysis section and in the corresponding Figures and Figure legends. Indeed, several significant changes were no longer significant after this correction, which is reflected in the description and discussion of the results. Nevertheless, we would like to emphasize that the n value for most plasma parameters in the experiment is relatively high (data from 9-11 animals per group) and, most important, that we trust the discussed results because they fit with known biological actions of exogenous ATRA treatment on different end-points (as elaborated in the Discussion).

- More information is needed regarding the FIA analysis, such as instrument conditions and sample preparation. The reference is insufficient.

The Reviewer’s questions and comments on methodological aspects led us contact again the experts who actually conducted in their laboratory and were in charge of the analyses, in the Institute of Physiology of the Czech Academy of Sciences, Prague, Czech Republic. In this process, a bulk  error we have been dragging during the writing of the original version of this Ms emerged, as the analysis were conducted  by liquid chromatography-tandem mass spectrometry (LC-MS/MS), not by FIA-ESI-MS/MS as written by mistake in the original version. Now more information is included in the Materials and Methods section regarding instrument conditions and sample preparation, and more specific methodological references are used. The persons in charge of the targeted metabolomic analysis, Ondrej Kuda and Jan Kopecky, have been incorporated as co-authors, in view of their involvement in the revision and previously in the experimental analyses.

- Does the kit include internal standards? Could data be presented as absolute concentrations instead of arbitrary units?

The kit includes deuterated internal standards only for certain acylcarnitines and amino acids. For other metabolites like DHA, ATP, etc, there are not standards and the usual way is to express them as arbitrary units or just normalized intensity. The same applies to gene expression data. We chose to use arbitrary units for all parameters tested because our aim is to compare the two experimental groups (ATRA and control) between them, rather than quantifying absolute levels.

- Without internal standards to control for differences in ionization efficiency between different metabolites and to control for differences in ion suppression between different samples, summing metabolite responses can be misleading. How is this dealt with here?

This question is more related to FIA, where the ion suppression is much bigger deal. As explained above, analyses were by LC-MS/MS, not FIA-ESI-MS/MS as written by mistake in the original version. In LC-MS/MS, the ion suppression and other negative effects are not that pronounced within metabolite families, and we only summed responses of metabolites within the same family (e.g. AA).

- It is well known that ATP can fragment into ADP and AMP in the MS source during ionization, and if not separated chromatographically, biological AMP can not be distinguished from the AMP signal that arises from ATP fragmentation in the source. How is this dealt with here?

As it was LC-MS/MS, the adenosine nucleotides were separated in time, thus the levels are not masked by non-specific or in source fragmentation.

- Metabolomics analysis of tissues involved in energy homeostasis (WAT, BAT, and liver) may be informative and provide answers to questions raised in the discussion.

We agree this would be interesting for future work, yet believe the actual Ms already stands by itself and is meaningful by itself.

Other changes introduced during the revision:
- Former Figure 2B has been omitted and the result is now instead described in the text, because composite parameters (such as GABR in former Figure 2A) have not been subjected to the Benjamini-Hochberg correction, and we felt the coexistence in a single figure of data corrected and not corrected complicated the legend unnecessarily.
- In the explanation of the Supplementary Materials, that of Supplementary Table 2 has been added, as it was involuntarily omitted by mistake in the original submission.
- Slight changes of redaction and a few typos (like cysteine for cystine) have been corrected.

Round 2

Reviewer 2 Report

The authors have made solid efforts to address my previous concerns. The manuscript is now more technically sound. After addressing the following minor comments, I recommend for publication.

Minor comments

-          Line 101: typo, p-value is listed as 0, “(P=0.000,”

-          Was a Student’s t-test used for plasma analysis, but Mann-Whitney used for PBMC analysis? If so, why?

Author Response

The authors have made solid efforts to address my previous concerns. The manuscript is now more technically sound. After addressing the following minor comments, I recommend for publication.

We appreciate the recognition of the revision work done raised by the Reviewer and thank them very much for their comments, as we believe the manuscript has gained by taking all the comments into account.

Minor comments

Line 101: typo, p-value is listed as 0, “(P=0.000,”

We changed to P<0.000.

Was a Student’s t-test used for plasma analysis, but Mann-Whitney used for PBMC analysis? If so, why?

Yes, we used Student’s t-test for plasma analysis and Mann-Whitney U test for PBMC analysis. We used the non-parametric statistical test when our data is not normally distributed and/or our sample size is relatively small as for PBMC analysis were the sample size per group was only 3-5.